# Simulating Dairy Herd Structure and Cash Flow: Design and Application of a Web-Based Decision-Support Tool

**DOI:** 10.3390/ani16010129

**Published:** 2026-01-02

**Authors:** Victor E. Cabrera

**Affiliations:** Department of Animal and Dairy Sciences, University of Wisconsin–Madison, Madison, WI 53706, USA; vcabrera@wisc.edu

**Keywords:** dairy farm management, herd dynamics, Markov chain model, economic simulation, replacement and reproduction decisions, scenario analysis, decision-support system, risk management, dairy management decision support tools (DairyMGT)

## Abstract

Dairy farmers regularly make long-term decisions about herd size, replacement, reproduction, and capital investments, often under tight cash-flow constraints. It is difficult to see how changes in culling rate, heifer rearing, calving interval, milk yield, or new loans will affect the numbers of animals in each age group and the money available to pay bills over time. The Dairy Herd Structure Simulation and Cash Flow tool is a web-based decision-support tool, available through the Dairy Management Decision Support Tools (DairyMGT.info) website, which helps producers explore these “what-if” questions before they commit real money. Users can either enter simple herd-level information or upload a spreadsheet with individual cow and heifer data. The tool then simulates herd structure and monthly cash flow up to 400 months into the future, accounting for milk income, feed, labor, heifer-raising costs, cull cow income, heifer purchases or sales, and loan amortization. The results are displayed as graphs and tables showing projected numbers of calves, heifers, and cows in each lactation, as well as income over variable cost per cow. By comparing alternative management strategies—including, but not limited to, herd expansion—the tool supports more informed planning and risk management for dairy farmers, consultants, and Extension professionals.

## 1. Introduction

Modern dairy farms are complex, dynamic enterprises in which biological processes, management choices, and market conditions interact over long time horizons. Decisions about culling, replacement, reproduction, and herd size affect not only short-term cash flow but also the future distribution of cows, heifers, and calves in the herd and, ultimately, whole-farm profitability.

In response to this complexity, a large suite of computerized decision-support tools has been developed to assist dairy producers and advisors in making data-driven economic and management decisions, particularly within the Dairy Management Decision Support Tools (DairyMGT.info) platform at the University of Wisconsin–Madison [1]. These tools typically focus on specific questions such as cow replacement value, heifer-raising costs, or ration optimization, and many are designed as relatively static calculators that compare alternative scenarios at a fixed point in time [1].

Whole-farm or system-level tools have more recently begun to couple herd dynamics with sustainability and economic metrics in user-friendly interfaces. For example, the DairyPrint model simulates herd demographics together with feed, manure, and greenhouse gas emissions to evaluate management and mitigation strategies at the farm scale [2]. These developments illustrate how dynamic simulation can be embedded in practical tools that help producers and other stakeholders explore “what-if” scenarios.

Parallel advances in integrated decision-support systems (IDSSs) and data integration infrastructures, such as the Dairy Brain initiative, aim to connect continuous on-farm data streams with analytical modules and dashboards, moving decision-support closer to real time and embedding it into daily management workflows [3,4,5]. These efforts highlight both the opportunity and the challenge: to translate increasingly rich data and sophisticated models into tools that are transparent, interpretable, and aligned with the practical questions producers face.

Within this broader decision-support landscape, Markov chain and dynamic programming methods have played a central role in the economic evaluation of dairy herd dynamics, especially in the context of replacement and reproductive decision-making [6,7,8]. Markov chain formulations have been used to compute cow value and the economic impact of pregnancy and pregnancy loss [6], to compare alternative reproductive programs combining timed artificial insemination and estrus detection [7], and to quantify how reproductive performance affects herd value when integrated with dynamic programming [8]. These studies provide important insights into long-run profitability and optimal decision rules at the cow level and have been summarized in recent reviews of mathematical methods for cow replacement problems [9]. However, they typically summarize results as steady-state herd performance or expected net returns under fixed policies, rather than presenting explicit time trajectories for herd structure and cash flow at the whole-farm level. This leaves a gap for tools that project both herd composition and cash-flow patterns over time in a way that can be directly used by farmers, consultants, and Extension professionals for day-to-day and strategic planning.

In practice, dairy producers face questions that are inherently transitional and time-bound, such as how a chosen expansion strategy will reshape herd structure over the next several years, how aggressive culling or changes in heifer-raising capacity will affect the supply of replacements, and how these biological dynamics will interact with cash flow, capital investments, and debt service at a monthly resolution. Existing tools rarely provide an integrated view of these herd-structure and financial trajectories in a form that remains intuitive for non-specialists.

At the same time, recent work on data integration and analytics in the dairy industry emphasizes the need for tools that are not only methodologically sound but also operationalizable in commercial environments, with clear inputs, interpretable outputs, and the ability to interface with other components of a broader decision-support ecosystem [1,2,3,4,5,10]. This context motivates decision-support tools that explicitly link herd structure, replacement and expansion strategies, and cash flow projections in a way that can be readily adopted by producers and consultants. Recent work comparing reproductive management strategies has also highlighted how such decisions translate into differences in costs and short-term cash flow at the herd level [11].

The objective of this paper is to describe the design and application of a web-based Dairy Herd Structure Simulation and Cash Flow decision-support tool. Specifically, we (i) present the underlying modeling framework, including the Markov chain representation of herd dynamics and the associated cash-flow calculations; (ii) describe the software implementation and user interface within the DairyMGT.info platform; and (iii) illustrate the tool’s use through two contrasting example scenarios. The first scenario represents a baseline steady-state herd in which a fixed target of 1000 adult cows is maintained by purchasing or selling heifers as needed, whereas the second scenario represents a heifers-driven herd in which replacements come exclusively from home-raised heifers and the herd is allowed to grow from a user-defined initial structure. By documenting both the model and its practical implementation, and by exemplifying these two types of management strategies, we aim to provide dairy producers, consultants, and Extension professionals with a transparent, reproducible, and accessible resource to support long-term herd planning and cash-flow risk management.

## 2. Materials and Methods

### 2.1. Decision Problem and Overall Design

The Dairy Herd Structure Simulation and Cash Flow tool was designed to support medium- and long-term planning decisions related to herd structure, replacement, reproduction, and capital investments on dairy farms. It focuses on questions such as how different culling and replacement strategies, reproductive performance levels, or expansion plans will affect the number of animals in each age and lactation group and the resulting monthly cash flow over time. The tool is implemented as a web-based application within the Dairy Management Decision Support Tools (DairyMGT.info) platform at the University of Wisconsin–Madison [1], with all computations performed on the server and results returned as interactive tables and graphs in the user’s web browser.

The model operates in discrete monthly time steps and can project up to 400 months into the future, allowing users to examine both transitional dynamics and long-run trajectories. The core of the herd-structure module is a Markov chain representation of transitions among animal states, similar in spirit to earlier applications in cow replacement and reproductive modeling [6,7,8], but adapted here to generate explicit herd inventories and to link these to detailed cash-flow calculations. The general design philosophy follows previous decision-support and integrated decision-support efforts that emphasize transparent biological assumptions, explicit economic components, and user-friendly interfaces for farm-level decision-making [1,2,3,4,5,10].

### 2.2. Herd Structure and Markov Chain Formulation

Animals are grouped into biologically and managerially relevant states. Cows are classified by parity (first and later lactations), months in milk, and pregnancy stage, while heifers are classified by age in months. Calves can be treated as a separate group or as the earliest heifer age class, depending on user-defined management. Within each monthly time step, animals can transition among these states according to probabilities that reflect calving, conception, pregnancy loss, culling, death, and aging, as is common in Markov chain representations of dairy herds [6,7,8].

Formally, the herd structure at month t is represented by a state vector $n_{t}$ whose elements correspond to the number of animals in each state (e.g., heifers of a given age, cows in a given parity–month-in-milk–pregnancy class). The evolution of $n_{t}$ follows$$n_{t+1}=P_{t}n_{t}+b_{t},$$
where $P_{t}$ is a transition probability matrix derived from user-specified rates (e.g., conception probability, voluntary waiting period, culling rate), and $b_{t}$ is a vector of exogenous entries such as purchased heifers or externally specified calf inflows. This structure mirrors previous Markov chain models for cow value and reproductive programs [6,7,8] but is simplified for monthly rather than daily time steps and oriented toward simulation rather than optimization, consistent with the broader family of replacement models reviewed in Palma et al. [9].

Initial conditions can be specified in two ways. Users may (i) enter a set of herd-level summary values, such as total cows by parity and average days in milk, from which the tool constructs an approximate starting state vector, or (ii) upload a spreadsheet with individual-animal data. The template file allows entry of cow identification, lactation number, and days in milk for each cow, as well as identification and age for each heifer. The uploaded file is parsed and aggregated into the internal state vector $n_{0}$, ensuring that the simulated herd closely reflects the current herd structure of a specific farm.

In addition, the tool offers a “From steady state” option. When this box is selected, the model iteratively applies the transition matrix until the distribution of animals among states changes negligibly from one month to the next, and the simulation horizon presented to the user begins at this approximate equilibrium. This mode represents herds that are already operating under long-run demographic conditions implied by the chosen biological parameters and is useful for exploring generic or “typical” herds. When the option is not selected, the simulation starts directly from the user-specified initial structure, allowing explicit study of transitional dynamics as the herd moves toward or away from equilibrium.

### 2.3. Economic and Cash-Flow Components

The economic module aggregates revenues and costs for each monthly time step by linking the simulated herd structure to user-defined economic parameters. The model is strictly cash-flow based and tracks monthly cash inflows and outflows; it does not explicitly model depreciation, taxes, or asset values. This approach is consistent with earlier decision-support tools that combine biological models with farm-level economic calculations to evaluate management strategies [1,2,6,7,8]. Key revenue components include milk income, calculated as the product of milk yield per cow (by parity and stage of lactation) and milk price, and income from the sale of cull cows and surplus heifers. Variable costs include feed and labor for each animal group, heifer-raising costs, and miscellaneous operating expenses. All prices (e.g., milk, feed, labor, cull cows, heifer purchases and sales) and most technical coefficients (e.g., feed offered per animal group, labor hours per animal) are provided by the user as editable inputs. Default values shown in the interface are intended only as generic placeholders and are not drawn from any external cost database. Users specify these parameters in the “Forecast Inputs” and “Herd Structure Simulation” tabs, where over 30 editable inputs capture herd-specific prices, quantities, and technical coefficients.

Milk production is modeled at the herd-average level. The user specifies a rolling herd average (RHA) for milk yield, and the tool internally maps this RHA to a set of standardized lactation curves for first, second, and third-and-later parities. These curves define expected milk yield per cow for each parity and month in milk. Monthly milk production is then computed as the sum, over all parity–month-in-milk classes, of the expected yield per cow in that class multiplied by the number of cows in that class. Dry matter intake (DMI) per cow in each group is predicted using standard requirement-type equations as a function of milk yield, body weight, parity, and stage of lactation, and feed costs are obtained by multiplying predicted DMI by the user-specified diet cost for each animal group. Feed consumption and labor requirements are similarly parameterized per animal group. The tool does not use genetic evaluation traits or individual cow records, nor does it rely on an external nutritional model to define milk production potential; all production expectations are derived from the user-specified herd-level RHA and the parity-specific lactation curves implemented in the tool.

Default mode (checkbox off). The adult herd is maintained at a user-specified target number of cows. Shortages of home-raised heifers are automatically compensated with purchased animals, whereas surplus heifers can be sold. In this mode, heifer purchases and sales act as balancing items that keep the milking herd close to its target size, and associated cash inflows and outflows are included in the monthly cash flow.Heifers-driven mode (checkbox on). The model assumes that only heifers raised on the farm enter the adult herd and that no replacements are purchased or sold. Adult herd size becomes an outcome of the simulated reproduction, mortality, culling, and heifer-rearing parameters rather than a fixed target. The resulting projections reveal whether the specified management and biological performance would, by themselves, cause the herd to gradually expand, remain stable, or shrink over the planning horizon. In addition, the interface includes fields for specifying the number of pregnant heifers (springers) purchased or sold at user-defined months, together with their purchase and sale prices. These transactions directly affect replacement flows and herd size and provide a flexible way to represent intermediate strategies in which only a proportion of heifers are retained or in which targeted expansions or reductions are implemented at specific times.

The tool allows users to specify additional investments and associated bank loans, including principal, interest rate, and amortization schedule. From these inputs, the model computes a constant monthly loan payment (including principal and interest) over the chosen repayment period, which is added to cash outflows as “Loan Amortization” for each month in which the loan is active. Loan payments are incorporated into monthly cash outflows. Because the focus is on cash flow, the underlying asset and its depreciation schedule are not modeled explicitly; any investment that can be represented by a loan with fixed monthly payments can be analyzed within this framework.

For each month, the tool computes a set of herd and economic outputs that are stored over the full planning horizon. The main economic outputs are Milk Income, Feed Costs, Labor Costs, Raising Heifers Costs, Heifer Purchases, Heifer Sales, Loan Amortization, income over variable cost (IOVC) per cow, IOVC for the herd, Cull Cow Income, and Miscellaneous Expenses. The main herd output is the Number of Cows, alongside the distribution of animals across heifer age classes and cow parity and months in milk. These monthly outcomes are available as tables and time-series plots within the interface and can be exported as spreadsheets, enabling readers to replicate the example scenarios by using the parameter values and initial conditions described in the Results section.

## 3. Results and Example Scenarios

To illustrate the use of the Dairy Herd Structure Simulation and Cash Flow tool, two example scenarios were constructed for a 1000-cow Holstein herd over a 120-month horizon. Both scenarios used the same biological and economic parameters (Figure 1), including a 24-month age at first calving, a 13-month calving interval, and a 35% annual cow replacement rate. The scenarios differ in how replacements are handled and in the initial conditions (steady state versus a user-defined starting structure). In Table 1, income over variable cost (IOVC) is reported as the mean total monthly value for the herd.

### 3.1. Scenario 1: Baseline Steady-State Herd with Target Size

In the baseline scenario, the “From steady state” option was activated and the number of adult cows was fixed at 1000. Replacement needs were automatically met by purchasing or selling heifers as required (the “Heifers → Cows” option was unchecked). Under these assumptions, the simulated herd structure was essentially constant over the 10-year projection (Figure 2A). The number of adult cows remained at 1000 throughout, and the heifer inventory showed only minor month-to-month variation.

Cash-flow outputs were correspondingly stable (Figure 3A, Table 1). In the baseline steady-state scenario, mean monthly milk income, feed costs, labor costs, raising-heifer costs, cull-cow income, and miscellaneous expenses were USD 338,589, 152,179, 81,588, 76,393, 25,839, and 16,667, respectively, yielding a mean monthly IOVC for the herd of USD 37,602 (Table 1). Monthly IOVC varied very little over the 120 months (minimum USD 37,550; maximum USD 37,664; CV = 0.30%; Table 1), consistent with the nearly constant herd size and production level. This scenario illustrates how the tool can characterize the long-run herd structure and cash-flow pattern of an established herd operating under current management.

### 3.2. Scenario 2: Heifers-Driven Herd Growth Without Steady State

The second scenario used the same biological and economic parameters but was designed to show how herd size and cash flow respond when replacements must come entirely from within the herd. In this case, the “From steady state” option was not selected, and the simulation started from a user-defined initial herd structure with 1000 adult cows and a corresponding heifer inventory. The “Heifers → Cows” option was activated so that all heifers raised on the farm entered the adult herd and no replacement animals were purchased or sold.

Under these conditions, the model predicted a marked transient followed by sustained herd growth (Figure 2B). Adult cow numbers fell to about 500 during the first two years as existing cows were culled faster than home-raised heifers could enter the milking herd, then increased steadily, reaching roughly 1356 cows by month 120. On average, the herd contained about 1143 cows over the 10-year period, 14% more than in the baseline scenario (Table 1). Milk income increased in proportion to the larger herd size, but raising-heifer (not shown), feed, and labor costs also increased and became more variable. IOVC remained positive but exhibited a wider range over time than in the steady-state scenario (Figure 3B, Table 1). This example shows how the tool can diagnose whether a given combination of reproductive performance, culling rate, and heifer-rearing strategy would, in the absence of purchases, cause the herd to shrink, remain stable, or grow, and how these dynamics translate into cash-flow risk. Consistent with these more pronounced herd dynamics, monthly IOVC showed much greater variability in the heifers-driven growth scenario than in the steady-state scenario (CV = 306% vs. 0.30%; Table 1), highlighting the tool’s ability to quantify cash-flow risk over time. In the heifers-driven herd growth scenario, mean monthly milk income and all major cost components were higher than in the baseline (USD 385,261 milk income; 173,171 feed; 92,834 labor; 78,842 raising-heifer costs; 29,759 cull-cow income; and 16,667 miscellaneous expenses), resulting in a higher mean monthly IOVC of USD 53,506 for the herd (Table 1). However, IOVC was much more variable over time, ranging from USD −54,578 to USD 109,284 with a coefficient of variation of 306% (Table 1), reflecting the transient nature of the expansion phase.

### 3.3. Other Applications

Beyond the two scenarios presented here, the same framework can be used to evaluate loan-financed expansions or infrastructure improvements by specifying loan amount, interest rate, amortization period, and start of payments. Loan amortization then appears as an additional monthly cost in the cash-flow outputs, allowing users to visualize how financed investments affect IOVC and net cash flow over time.

## 4. Discussion

This Technical Note describes a web-based decision-support tool that combines a Markov chain representation of dairy herd dynamics with a monthly cash-flow model in a form that is directly usable by producers, consultants, and Extension professionals. The Dairy Herd Structure Simulation and Cash Flow tool builds on previous decision-support efforts within the DairyMGT.info platform [1] and complements whole-farm sustainability tools such as DairyPrint [2] and emerging integrated decision-support systems and data-integration infrastructures, including the Dairy Brain initiative [3,4,5,10]. By explicitly linking herd structure, replacement strategies, and cash flow in monthly time steps over long planning horizons, the tool is designed to fill a gap between research-oriented herd models [6,7,8,9] and the practical, scenario-based questions that producers face.

Compared with other decision-support tools described in the literature [1,2,4,9,10], the Dairy Herd Structure Simulation and Cash Flow tool occupies a distinct niche. Many academic models for replacement or reproductive decision-making focus on steady-state herd performance or optimal policies under long-run economic criteria [6,7,8], and their implementations are often not directly exposed to producers as interactive tools. Commercial herd-management software and on-farm dashboards typically provide detailed reports on current inventories, production, and cash flows but do not explicitly project herd structure and cash-flow trajectories months or years into the future under alternative assumptions. By contrast, the present tool combines a Markov chain herd model with a monthly cash-flow calculator in a single interface, allowing users to start either from a demographic steady state or from their actual herd, to switch between purchased and home-raised replacement strategies, and to visualize the resulting herd structure and cash-flow patterns over long planning horizons. In this way, it complements both research-oriented optimization models and more conventional management software by focusing explicitly on forward-looking, scenario-based analysis.

The contrasting example scenarios presented in this Technical Note, a baseline steady-state herd and a heifers-driven herd growth scenario, illustrate this differential by showing how long transitional periods in herd size translate into marked differences in monthly cash-flow patterns and IOVC variability, which would be difficult to obtain from steady-state economic models or static accounting dashboards.

The example scenarios highlight several ways in which this type of tool can support decision-making by dairy producers, consultants, and Extension agents. In the steady-state scenario, the tool provides a transparent description of a “business-as-usual” herd, clarifying the implicit steady-state herd structure implied by a given combination of reproduction, culling and heifer-rearing parameters and translating this into a smooth cash-flow trajectory. Such baseline simulations are useful for benchmarking current performance, communicating with lenders and advisors, and setting expectations before contemplating changes in management or scale. In practice, such baseline simulations are useful when preparing five-to-ten-year plans for herd expansion or contraction and when evaluating whether current reproductive and replacement performance is sufficient to sustain a desired herd size without large swings in cow numbers. When the “Heifers → Cows” option is engaged and the model is not started from steady state, the second scenario shows how the same biological parameters can lead to substantial transient behavior and long-term growth when the herd is forced to rely solely on internally raised replacements. The initial decline in cow numbers, followed by gradual expansion beyond the original herd size, underscores how replacement strategies affect not only long-run capacity but also the timing of cash surpluses and deficits.

These scenarios also demonstrate the value of explicitly tracking IOVC and other cash-flow measures at a monthly resolution. For example, Scenario 2 shows that relying exclusively on home-raised replacements can increase long-run milk income and IOVC but may expose the farm to periods of lower cash availability while the replacement pipeline is being built. The ability to visualize such patterns is particularly important when planning herd expansions, adjusting reproductive programs, or considering changes in culling policies. Because the tool reports monthly cash-flow components and IOVC, it can also provide a common framework for producers and lenders to examine alternative financing plans, for example, by comparing how different replacement strategies or investment schedules affect short-term liquidity versus long-term profitability. Although loan-financed investments were not explored in detail here, the model’s loan module offers a straightforward way to examine how debt service interacts with these biological dynamics, which is a recurring concern in conversations between producers and financial institutions. Beyond the two illustrative replacement modes, producers can also use the springer purchase and sale options to model intermediate strategies, such as retaining only a proportion of home-raised heifers while periodically buying or selling pregnant heifers to fine-tune herd growth or contraction.

From a validation perspective, the tool was primarily designed to be internally consistent with established reproductive, culling, and replacement relationships reported in the literature [6,7,8,9]. In Extension workshops and one-on-one consultations, model parameters have been adjusted to reflect specific herds using available DHIA and accounting records, and the resulting trajectories for herd size, milk sold, and replacement flows have been broadly consistent with producers’ expectations and observed aggregate patterns. A formal validation against longitudinal datasets is beyond the scope of this Technical Note, but it is a priority for future work. The current implementation allows users to explore parameter uncertainty through manual scenario analysis (e.g., varying conception rate, replacement rate, or heifer-raising costs), and future versions could include built-in sensitivity analysis tools as well as stochastic extensions in which key parameters are treated as random variables. Such developments would allow the tool to provide not only point forecasts but also ranges or probability bands for herd structure and cash-flow outcomes, further strengthening its value for risk assessment and planning.

Like any model-based tool, the Dairy Herd Structure Simulation and Cash Flow tool has limitations that should be recognized when interpreting its outputs. The current implementation is deterministic and operates at a monthly time step; it does not capture day-to-day variability in production, price volatility, or stochastic health events, nor does it quantify uncertainty in user-specified parameters. The model does not impose explicit constraints on facilities, labor availability, or environmental regulations, so users must judge whether simulated herd sizes and cash flows are feasible in their specific context. Furthermore, the tool is designed for scenario analysis rather than optimization: it does not search for “optimal” policies but instead provides a consistent framework for comparing user-defined alternatives. Market-driven variables such as milk, feed, and cull-cow prices are treated as user-specified constants within each scenario, so the model does not explicitly represent price volatility. Uncertainty in these prices is intended to be explored through scenario analysis (for example, by comparing simulations under low, average, and high price assumptions) rather than through an assumed stochastic price process. Future versions of the tool could incorporate built-in sensitivity analysis and stochastic price trajectories to provide users with explicit ranges or probability bands for cash-flow outcomes. In addition, the model operates at the herd-average level and does not track individual cows or their genetic merit. Culling and heifer selection are represented through user-specified culling rates, reproductive performance, and replacement flows, rather than through estimated breeding values or genomic selection indices. In practice, users can only approximate the consequences of alternative genetic strategies by modifying parameters such as RHA milk yield, reproductive performance, and longevity, so explicit integration of genetic information remains an important area for future development.

Despite these limitations, the tool offers several advantages for Extension and teaching. Its web-based implementation, parameter-driven structure and graphical outputs make it suitable for workshops, classroom exercises, and one-on-one consultations, where users can quickly explore “what-if” questions and see the consequences of different assumptions. The same Markov chain and economic framework can also serve as a building block for more advanced applications, including integration into data-driven IDSS architectures [3,4,5,10] or coupling with optimization and reinforcement-learning approaches to suggest candidate management strategies. Future work could focus on tighter integration with farm data streams, validation with longitudinal herd records and expansion of the interface to support batch scenario runs and automated reporting.

## 5. Conclusions

The Dairy Herd Structure Simulation and Cash Flow tool provides a practical, web-based implementation of a Markov chain herd model linked to a monthly cash-flow calculator for dairy farms, designed primarily for use by dairy producers, consultants, and Extension professionals. By allowing users to specify herd structure, reproduction and culling parameters, heifer-rearing strategies, and economic conditions, and then projecting herd composition and cash flows up to 400 months into the future, the tool offers an accessible way to explore the medium- and long-term consequences of management and investment decisions.

The example scenarios presented here demonstrate how the tool can represent both stable, steady-state herds and transitional dynamics when herds rely solely on internally raised replacements. Differences in herd size, milk income, and IOVC between scenarios illustrate how replacement strategies shape both biological trajectories and financial risk. Although the model is simplified and intended for scenario analysis rather than precise prediction, it can support producers, consultants, and Extension professionals in benchmarking current performance, evaluating alternative herd-structure strategies and preparing for expansion or other major changes in herd management, and supporting discussions with lenders by presenting projected monthly cash-flow and IOVC patterns under alternative replacement and financing strategies.

## Figures and Tables

**Figure 1 animals-16-00129-f001:**
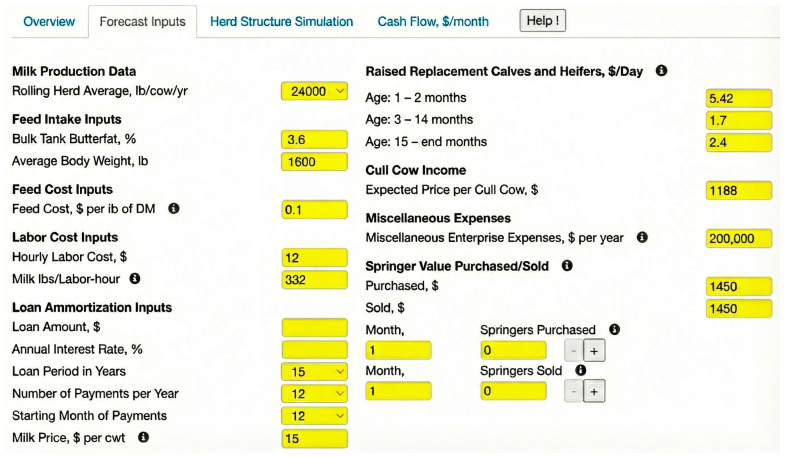
Input parameters for the example herd. Forecast input screen of the Dairy Herd Structure Simulation and Cash Flow tool for the 1000-cow Holstein herd, showing key biological and economic parameters (rolling herd average milk yield, feed and labor costs, heifer-rearing costs, cull cow value, miscellaneous expenses, and milk price), as well as loan amortization fields.

**Figure 2 animals-16-00129-f002:**
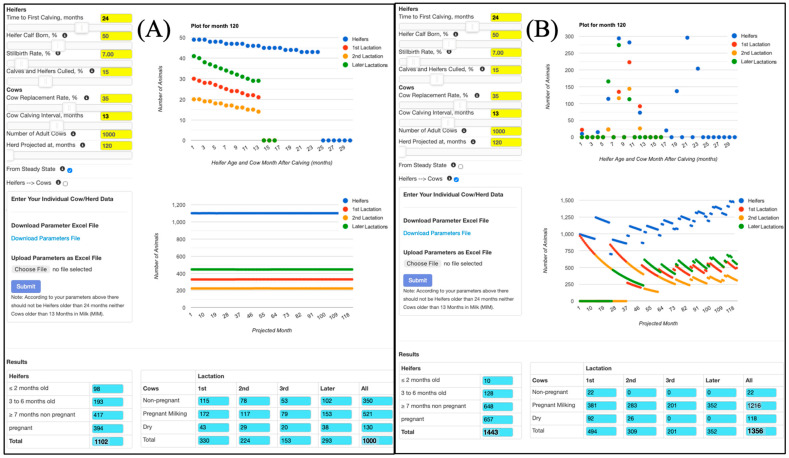
Simulated herd structure over 120 months for two replacement strategies. (**A**) Scenario 1: baseline steady-state herd with a fixed target of 1000 adult cows (“From steady state” option selected; “Heifers → Cows” unchecked), showing essentially constant numbers of heifers and cows by parity and the final herd inventory. (**B**) Scenario 2: heifers-driven herd growth starting from a user-defined initial structure (“From steady state” not selected; “Heifers → Cows” checked), illustrating the initial decline and subsequent expansion in cow numbers and the build-up of older heifer age classes over the 120-month horizon. In both panels, blue symbols represent heifers, orange symbols cows in first lactation, green symbols cows in second lactation, and red symbols cows in third and later lactations.

**Figure 3 animals-16-00129-f003:**
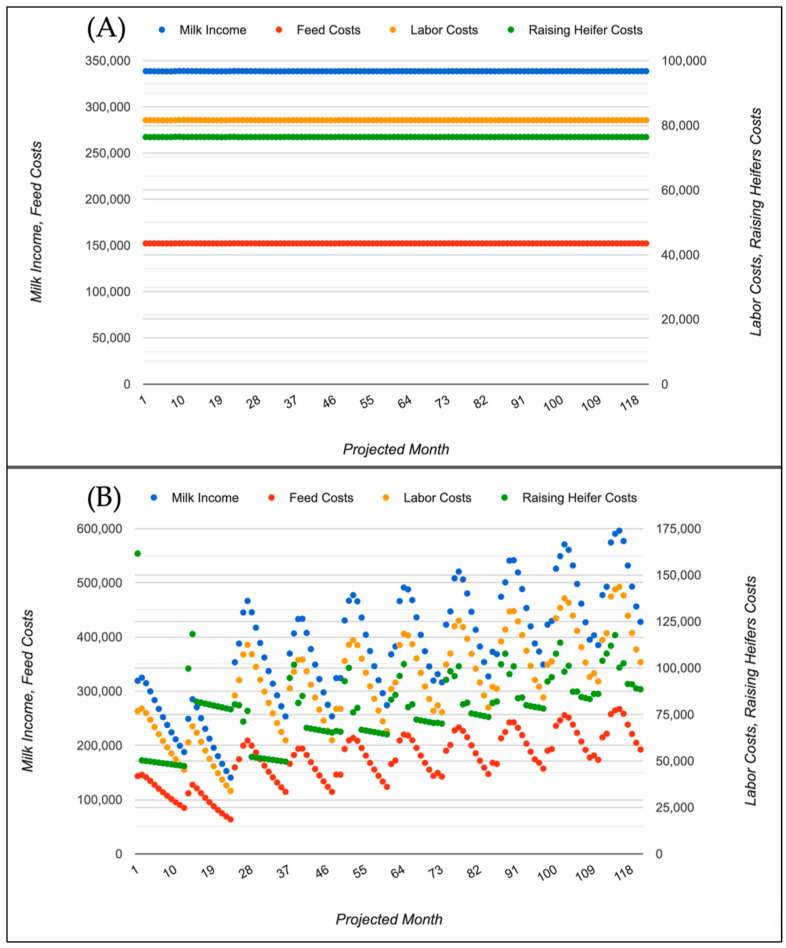
Monthly cash-flow most critical components and income over variable cost (IOVC) per herd for the two scenarios. (**A**) Scenario 1: baseline steady-state herd, showing nearly constant monthly milk income, feed, labor and heifer-raising costs, and relatively stable IOVC. (**B**) Scenario 2: heifers-driven herd growth, showing larger variation and upward trends in milk income and cost components, and a wider range of IOVC reflecting the transient dynamics and subsequent expansion of the herd. IOVC = milk income − feed costs − labor costs − raising heifer costs + cull cow income − miscellaneous expenses − heifer purchases + heifer sales − loan amortization.

**Table 1 animals-16-00129-t001:** Summary statistics from the two example scenarios for a 1000-cow Holstein herd simulated over 120 months.

Scenario	Cows, Month 1	Cows, Month 120	Mean Cows	IOVC ^1^, Month 120 (USD/Month)	Min IOVC (USD/Month)	Max IOVC (USD/Month)	CV of Monthly IOVC (%)
Baseline steady-state herd	1000	1000	1000	37,601	37,550	37,664	0.30
Heifers-driven herd growth	971	1356	1143	62,618	−54,578	109,284	306
**120-month mean economic outcomes ^2^** ** (USD/month)**							
**Scenario**	**Milk income**	**Feed** **costs**	**Labor** **costs**	**Raising** **heifer costs**	**Cull cow** **income**	**Miscellaneous expenses**	**IOVC**
Baseline steady-state herd	338,589	152,179	81,588	76,393	25,839	16,667	37,602
Heifers-driven herd growth	385,261	173,171	92,834	78,842	29,759	16,667	53,506

^1^ Income over variable cost (IOVC) = milk income − feed costs − labor costs − raising heifer costs + cull cow income − miscellaneous expenses − heifer purchases + heifer sales − loan amortization. ^2^ Heifer purchases, heifer sales, and loan amortization were zero in all months for both analyzed scenarios.

## Data Availability

No new data were created or analyzed in this study. The simulation outputs shown in the example scenarios can be reproduced using the Dairy Herd Structure Simulation and Cash Flow tool available at DairyMGT.info (https://dairymgt.info, accessed on 26 December 2025).

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
