# Peer review of "Simulating Dairy Herd Structure and Cash Flow: Design and Application of a Web-Based Decision-Support Tool"

_animals, 2026, doi:10.3390/ani16010129_

Round 1
Reviewer 1 Report
Comments and Suggestions for Authors
General comments:
This technical note describes a web-based Markov-chain simulator designed as a decision-support tool for dairy herd management. It uses user-specified transition probabilities and parameters to project monthly herd structure (cows, heifers, calves) and cash flows up to 400 months.
The manuscript is within the journal’s scope, and the innovation is adequate for a Technical Note, provided that the authors present a novelty argument, contextualize their contribution more explicitly relative to existing tools, and slightly expand on validation and limitations. The manuscript does not propose a fundamentally new mathematical method, but it makes an original contribution at the decision-support and implementation level, since many prior tools treat herd dynamics and financial planning separately; here, biological and economic trajectories are presented together as time series, which is very relevant for farm planning and risk analysis.
Overall, the tool description is technically convincing and transparent at the level suited to a Technical Note, though a little more emphasis on empirical grounding would further strengthen acceptance prospects. Please check my suggestions below, regarding the format and clarity of presentation.
Suggested review
- Figure 1 (input parameters screen) is helpful to show interface design and parameter richness; ensure the resolution remains adequate and fonts are readable in the final layout.
- Figure 2 contrasts herd structure trajectories under the two scenarios. Consider adding very brief captions within the panels or improving the legend to guide readers (e.g., explicitly stating that blue symbols are cows and orange and green are heifer classes).
- Table 1 is succinct and essential; it might be helpful to add 1–2 simple risk metrics (e.g., min/max IOVC or coefficient of variation) if easily available from the tool, to emphasize risk-management capabilities.
- In the Discussion and Conclusions, emphasize a little more that the primary users are dairy producers, consultants, and Extension agents, and that the tool is particularly suitable for: planning herd expansion or contraction; exploring different replacement strategies (purchase vs. home-raised heifers); and discussing financing plans with lenders by showing monthly IOVC and cash-flow patterns.
- Add a short paragraph indicating how the tool could be (or has been) compared to real farm data (even if only at an aggregate level) and how parameter uncertainty might be addressed in future versions (e.g., sensitivity analysis, stochastic extensions).
My recommendation is a minor revision.
Author Response
Comment 1:
General comments:
This technical note describes a web-based Markov-chain simulator designed as a decision-support tool for dairy herd management. It uses user-specified transition probabilities and parameters to project monthly herd structure (cows, heifers, calves) and cash flows up to 400 months.
The manuscript is within the journal’s scope, and the innovation is adequate for a Technical Note, provided that the authors present a novelty argument, contextualize their contribution more explicitly relative to existing tools, and slightly expand on validation and limitations. The manuscript does not propose a fundamentally new mathematical method, but it makes an original contribution at the decision-support and implementation level, since many prior tools treat herd dynamics and financial planning separately; here, biological and economic trajectories are presented together as time series, which is very relevant for farm planning and risk analysis.
Overall, the tool description is technically convincing and transparent at the level suited to a Technical Note, though a little more emphasis on empirical grounding would further strengthen acceptance prospects. Please check my suggestions below, regarding the format and clarity of presentation.
Response 1: I thank the reviewer for the careful reading of the manuscript and for the very positive overall assessment. I appreciate the recognition that the main contribution of this Technical Note lies in the integration of herd dynamics and financial trajectories in a web-based decision-support tool. I fully agree with the reviewer’s suggestions to more clearly articulate the novelty relative to existing tools, to better contextualize the contribution within the current decision-support landscape, and to expand the discussion of validation and limitations. I have revised the manuscript accordingly and address each of the specific comments in detail below.
Suggested review
Comment 2:
- Figure 1 (input parameters screen) is helpful to show interface design and parameter richness; ensure the resolution remains adequate and fonts are readable in the final layout.
Response 2: I thank the reviewer for this comment. I agree that the readability of the interface screenshot is important. Figure 1 has been replaced with a higher-resolution image (≥300 dpi), and I have checked that all fonts and labels remain clearly legible in the PDF and should be readable in the final layout. Based on this comment, we have also reviewed other figures and improved their overall quality.
Comment 3:
- Figure 2 contrasts herd structure trajectories under the two scenarios. Consider adding very brief captions within the panels or improving the legend to guide readers (e.g., explicitly stating that blue symbols are cows and orange and green are heifer classes).
Response 3: I appreciate the suggestion to improve the guidance in Figure 2. I have revised Figure 2 to present the two scenarios side-by-side with clear panel labels (A) and (B) and maintained the legends within each panel. I also updated the figure caption to explicitly describe the color coding (blue = heifers; orange, green, and red = cows in successive lactations). These changes should make the trajectories and group distinctions clearer to readers.
Comment 4:
- Table 1 is succinct and essential; it might be helpful to add 1–2 simple risk metrics (e.g., min/max IOVC or coefficient of variation) if easily available from the tool, to emphasize risk-management capabilities.
Response 4: I thank the reviewer for this helpful suggestion. Although the model is deterministic, the monthly variation in IOVC over the simulated horizon is directly relevant to cash-flow risk. In response, I have expanded Table 1 to include simple variability metrics for IOVC (minimum, maximum, and coefficient of variation of monthly IOVC over the 120 months) for each scenario together with text highlighting the contrast to emphasize the tool’s usefulness for risk-management analysis. The table now also includes detailed outcomes of economic outcomes such as milk income, feed cost, etc. for both analyzed scenarios.
Comment 5:
- In the Discussion and Conclusions, emphasize a little more that the primary users are dairy producers, consultants, and Extension agents, and that the tool is particularly suitable for: planning herd expansion or contraction; exploring different replacement strategies (purchase vs. home-raised heifers); and discussing financing plans with lenders by showing monthly IOVC and cash-flow patterns.
Response 5: I appreciate the reviewer’s suggestion. I agree that the intended primary users and practical use cases should be emphasized more clearly. I have revised the Discussion and Conclusions to explicitly highlight dairy producers, consultants, and Extension agents as the main audiences for the tool (375-376), and to state that it is particularly suitable for planning herd expansion or contraction, exploring alternative replacement strategies (purchased vs. home-raised heifers), and supporting discussions with lenders by visualizing monthly IOVC and cash-flow patterns (382-385, 398-401).
Comment 6:
- Add a short paragraph indicating how the tool could be (or has been) compared to real farm data (even if only at an aggregate level) and how parameter uncertainty might be addressed in future versions (e.g., sensitivity analysis, stochastic extensions).
Response 6: I appreciate the reviewer’s suggestion to clarify how the tool relates to real farm data and how parameter uncertainty could be handled. I have added a paragraph in the Discussion describing (i) how the model has been compared qualitatively with farm records in Extension and consulting settings, and (ii) how future developments could incorporate more formal validation, systematic sensitivity analysis, and stochastic extensions to represent parameter uncertainty (408-421).
Comment 7:
My recommendation is a minor revision.
Response 7: I thank your positive feedback and appreciation of my work very much.
Reviewer 2 Report
Comments and Suggestions for Authors
Dear Author,
Congratulations on your technical note. Ways to improve management in farm animals are always welcome in the sector, as they make the lives of farmers, consultants, and veterinarians easier. The tool is a real help for all these categories and could, in the future, be promoted as a reference tool. The technical note is well written, easy to understand, and has a logical flow. Some minor suggestions are listed below.
Lines 47-51 The paragraph provides a good introduction to the article; however, it needs to be supported by appropriate references for both short-term cash flow and the future distribution of cows. I let you here 2 articles that could be cited to support cash flow expenses (https://doi.org/10.3390/ani15162380; https://doi.org/10.3389/fvets.2023.1167387).
Lines 97–108: The paragraph is more a description of the tool, and the aim of the study is only briefly described. Please enlarge the aim and the objectives of the study here by stating that exemplifications of two types of management were used. This will help readers have greater interest in the study.
The tool is well described in the Materials and Methods section, and both scenarios are useful for understanding the usefulness of the tool. However, it is necessary in the Discussion section to briefly describe and compare the tool with others available on the market. The author described in an easy understandable way the limitations of the tool and possible ways to improve it.
Author Response
Comment 1:
Dear Author,
Congratulations on your technical note. Ways to improve management in farm animals are always welcome in the sector, as they make the lives of farmers, consultants, and veterinarians easier. The tool is a real help for all these categories and could, in the future, be promoted as a reference tool. The technical note is well written, easy to understand, and has a logical flow. Some minor suggestions are listed below.
Response 1: I thank the reviewer for the very positive and encouraging comments. I appreciate the recognition that the tool can support farmers, consultants, and veterinarians and that it has potential to become a reference resource for herd-structure and cash-flow planning. I am grateful for the reviewer’s assessment that the technical note is well written, easy to understand, and logically structured. I address the minor suggestions listed below point by point and have revised the manuscript accordingly.
Comment 2:
Lines 47-51 The paragraph provides a good introduction to the article; however, it needs to be supported by appropriate references for both short-term cash flow and the future distribution of cows. I let you here 2 articles that could be cited to support cash flow expenses (https://doi.org/10.3390/ani15162380; https://doi.org/10.3389/fvets.2023.1167387).
Response 2: I thank the reviewer for this helpful suggestion and for providing specific references. I agree that the Introduction should be supported by literature documenting the cash-flow implications of management decisions. To maintain the existing numbering of references, I have added a citation to Berean et al. (2025) [11] later in the Introduction where cash-flow considerations are discussed explicitly. This new reference reinforces the point that reproductive and replacement strategies have important short-term cash-flow and cost implications (117).
Comment 3:
Lines 97–108: The paragraph is more a description of the tool, and the aim of the study is only briefly described. Please enlarge the aim and the objectives of the study here by stating that exemplifications of two types of management were used. This will help readers have greater interest in the study.
Response 3: I thank the reviewer for this suggestion. I agree that the aim and objectives should be stated more explicitly and that briefly describing the two illustrative management strategies in the Introduction will help engage readers. I have therefore revised the final paragraph of the Introduction to expand the objective statement and to specify that two contrasting management scenarios (a steady-state herd with a fixed target size and purchased/sold heifers, and a heifers-driven herd relying solely on home-raised replacements from a user-defined initial structure) are used as examples (118-131).
Comment 4:
The tool is well described in the Materials and Methods section, and both scenarios are useful for understanding the usefulness of the tool. However, it is necessary in the Discussion section to briefly describe and compare the tool with others available on the market. The author described in an easy understandable way the limitations of the tool and possible ways to improve it.
Response 4: I thank the reviewer for this constructive suggestion. I agree that briefly positioning the tool relative to other available decision-support tools will help readers better understand its niche and added value. I have therefore added a paragraph in the Discussion that compares the Dairy Herd Structure Simulation and Cash Flow tool with other academic and commercial decision-support tools described in the literature, emphasizing that many existing tools treat herd dynamics and financial planning separately, whereas the present tool integrates a Markov-chain herd model with monthly cash-flow projections in a web-based, Extension-oriented format (355-369).
Reviewer 3 Report
Comments and Suggestions for Authors
Introduction
- There is no description of the state of the art and gap in knowledge in terms of economic evaluation of dairy herd dynamics. This topic is first mentioned in line 86 when the term "cash flow" is used to introduce the need of farmers.
M&M
- This section is very poor and insufficient. Given this is a science article , the experiment needs to be replicable. For the reader to be able to judge how the DSS operates the paper needs to show as best as possible all the assumptions and restrictions utilised. Only a high level description of the herd inputs and scenario restrictions is provided. The economic model is particularly lacking information on what type of costs are being considered, what is the source of costing where data is not provided by the user, what type of investments are considered, their respective depreciation, repayments periods, etc, etc.
- In line 174 he explains that "milk production is modeled as expected yield per cow for a given parity and month". It is not clear what is modeling using as inputs, particularly regarding mik production potential: is it EBVs (not menitoned), previous milk production (for multiparous) or nutrition modelling (e.g. NRDC)?
Results
- Why are the user-defined options only steady state and Heifer-cow? It could be very likely that the user wants to take and intermediate option, keeping only a proportion of heifers, hence adopting a different growth rate. Any reason why not to provide with this user-defined option?
- Why there is no stochasticity included in the markets driven variables (particularly price of milk, feed and culled cows? This would reflect much better the reality of long term predictions and provide a more solid outcome for decision making to the user.
- Table 1 is too lean, it will be important for the reader to understand the different components of cost and income, not just milk, feed and total IOVC.
Discussion
- There is no explicit discussion of the literature: what has been found before (what capacities have shown other similar tools) and what is the differential of this tool, as shwon by the results.
- Culling decision and heifer selection decision are normally based on genetic potential. This can be informed by EBVs or previous produciton Herd Recording infromation. Even in a simulated (theoretical) exercise, this practical aspect should have been considered. This limitation should be mentioned in the discussion
Author Response
Comment 1:
Introduction
- There is no description of the state of the art and gap in knowledge in terms of economic evaluation of dairy herd dynamics. This topic is first mentioned in line 86 when the term "cash flow" is used to introduce the need of farmers.
Response 1: I thank the reviewer for this insightful comment. I agree that the Introduction should more clearly describe the state of the art and the remaining gap in the economic evaluation of dairy herd dynamics. To address this, I have revised the two consecutive paragraphs that discuss Markov-chain and dynamic programming approaches [6–9] and the needs of producers (89-108). The first now explicitly summarizes how these models have been used for economic evaluation of replacement and reproductive decisions and clarifies that most of this work focuses on steady-state performance or long-run net returns rather than on explicit time trajectories for herd structure and cash flow at the whole-farm level. The following paragraph then emphasizes that producers face inherently transitional and time-bound questions and that existing tools rarely provide an integrated, intuitive view of herd-structure and financial trajectories. Together, these revisions more clearly motivate the development of the Dairy Herd Structure Simulation and Cash Flow tool, which projects both herd composition and monthly cash-flow patterns in a form suitable for use by farmers, consultants, and Extension professionals.
Comment 2:
M&M
- This section is very poor and insufficient. Given this is a science article , the experiment needs to be replicable. For the reader to be able to judge how the DSS operates the paper needs to show as best as possible all the assumptions and restrictions utilised. Only a high level description of the herd inputs and scenario restrictions is provided. The economic model is particularly lacking information on what type of costs are being considered, what is the source of costing where data is not provided by the user, what type of investments are considered, their respective depreciation, repayments periods, etc, etc.
Response 2: I thank the reviewer for this important comment. I agree that readers should be able to clearly understand the economic assumptions and the set of variables calculated by the tool so that the example scenarios can be replicated. Within the level of detail appropriate for a Technical Note, I have expanded Section 2.3 (Economic and Cash-Flow Components) to (i) specify that the model is strictly cash-flow based (without explicit depreciation or tax modelling), (ii) clarify that all prices and technical coefficients are provided by the user, with default values serving only as editable placeholders and not coming from an external database, (iii) describe more explicitly how loans and investments are represented, and (iv) list the main monthly economic and herd variables computed by the tool (Milk Income, Feed Costs, Labor Costs, Raising Heifers Costs, Heifer Purchases, Heifer Sales, Loan Amortization, IOVC per cow, IOVC for the herd, Cull Cow Income, Number of Cows, Miscellaneous Expenses). These revisions make the assumptions and outputs of the economic module more explicit and improve the replicability and interpretability of the decision-support tool.
Comment 3:
- In line 174 he explains that "milk production is modeled as expected yield per cow for a given parity and month". It is not clear what is modeling using as inputs, particularly regarding mik production potential: is it EBVs (not menitoned), previous milk production (for multiparous) or nutrition modelling (e.g. NRDC)?
Response 3: I thank the reviewer for raising this point. The original text was too brief and could give the impression that the model uses EBVs, previous individual records, or a full nutritional model (e.g., NRC) to define milk production potential, which is not the case. In the revised manuscript I clarify that milk production is modeled at the herd-average level using the user-specified rolling herd average (RHA), which is mapped internally to standardized parity-specific lactation curves. Monthly milk production is calculated as the sum of expected yield per cow in each parity–month-in-milk class multiplied by the number of cows in that class. I also explain that dry matter intake per cow is predicted using standard requirement-type equations as a function of milk yield, body weight, parity, and stage of lactation, and that feed costs are obtained by multiplying predicted intake by user-specified diet costs. No genetic evaluation traits or individual cow records are used to drive production potential. These additions to Section 2.3 make the production and intake assumptions more explicit while keeping the level of detail appropriate for a Technical Note.
Comment 4:
Results
- Why are the user-defined options only steady state and Heifer-cow? It could be very likely that the user wants to take and intermediate option, keeping only a proportion of heifers, hence adopting a different growth rate. Any reason why not to provide with this user-defined option?
Response 4: I thank the reviewer for this thoughtful comment. The current implementation indeed provides two main herd-structure modes: (i) a steady-state mode in which a fixed target number of cows is maintained by purchasing or selling heifers as needed, and (ii) a heifers-driven mode in which all home-raised heifers enter the herd and no animals are purchased or sold. These two modes were chosen as clear “end points” for users. However, the tool also includes an explicit option to purchase or sell pregnant heifers (springers) at user-defined months, which allows users to represent intermediate strategies and to accelerate or slow herd growth beyond the two extreme modes. In the revised manuscript I clarify the existence of this springer purchase/sale option in the Materials and Methods (231-236) and briefly mention in the Discussion that it can be used to represent partial retention strategies without changing the core replacement mode (404-407). The example scenarios in this Technical Note do not vary springer purchases or sales, but the functionality is available to users for more detailed planning.
Comment 5:
- Why there is no stochasticity included in the markets driven variables (particularly price of milk, feed and culled cows? This would reflect much better the reality of long term predictions and provide a more solid outcome for decision making to the user.
Response 5: I thank the reviewer for this insightful comment. I agree that uncertainty in market-driven variables such as milk, feed, and cull-cow prices is an important aspect of long-term decision making. In the current version of the tool, these prices are user-specified constants for each scenario, and the model is fully deterministic. This design choice was made to keep the tool transparent and easy to explain in Extension and consulting settings, and to avoid imposing a particular stochastic price process that might not reflect the conditions of a given region or time period. Users are encouraged to explore price uncertainty through scenario analysis (e.g., running separate simulations under “low,” “average,” and “high” price assumptions).
To acknowledge this limitation more clearly, I have expanded the Discussion to state that market prices are treated deterministically in the present implementation and that incorporating stochastic price dynamics or built-in sensitivity analysis for key price variables is a natural direction for future development of the tool (432-439).
Comment 6:
- Table 1 is too lean, it will be important for the reader to understand the different components of cost and income, not just milk, feed and total IOVC.
Response 6: I thank the reviewer for this useful suggestion. I agree that showing additional components of costs and income helps readers better understand how the two scenarios differ economically. In the revised manuscript, Table 1 has been expanded and reorganized into two parts. The upper part summarizes herd size and IOVC behavior (IOVC in month 120, minimum and maximum monthly IOVC over the 120 months, and the coefficient of variation of monthly IOVC) for each scenario. The lower part reports 120-month mean monthly cash-flow components, including milk income, feed costs, labor costs, raising-heifer costs, cull-cow income, miscellaneous expenses, and mean IOVC. Footnotes clarify the IOVC definition and indicate that heifer purchases, heifer sales, and loan amortization were zero in all months for the analyzed scenarios. The Results section has been updated to briefly interpret these additional components. Together with the time-series plots in Figure 3, the revised table now provides a more complete yet still concise summary of the economic structure of the two example scenarios.
Comment 7:
Discussion
- There is no explicit discussion of the literature: what has been found before (what capacities have shown other similar tools) and what is the differential of this tool, as shwon by the results.
Response 7: I thank the reviewer for this helpful comment. I agree that the Discussion should more explicitly relate the findings to previous literature on decision-support tools and clearly state the differential contribution of the present tool. To address this, I have expanded the Discussion paragraph that compares the Dairy Herd Structure Simulation and Cash Flow tool with other models and software [1,2,4,6–10] so that it (i) summarizes the main capacities of existing academic and commercial tools, and (ii) explicitly highlights how the integrated herd-structure and cash-flow trajectories demonstrated in the two example scenarios differ from what those tools typically provide. The added sentences emphasize that the model’s ability to represent long transitional periods and IOVC variability under expansion is a key distinguishing feature relative to steady-state economic models and static accounting dashboards (335-369).
Comment 8:
- Culling decision and heifer selection decision are normally based on genetic potential. This can be informed by EBVs or previous produciton Herd Recording infromation. Even in a simulated (theoretical) exercise, this practical aspect should have been considered. This limitation should be mentioned in the discussion
Response 8: I thank the reviewer for this important observation. I agree that, in practice, culling and heifer selection decisions are strongly influenced by genetic potential, typically informed by estimated breeding values or by previous herd-recording information. The present tool operates at the herd-average level and does not track individual cows or their genetic merit, so genetic selection cannot be modeled explicitly. In the revised manuscript, I have added a few sentences in the Discussion noting this as a limitation and clarifying that genetic strategies can only be represented indirectly (e.g., through changes in rolling herd average, reproductive performance, and longevity parameters) in the current implementation. I also note that adding an explicit genetic component would be a valuable extension for future versions of the tool (439-446).